# Breaking Speed Limits with Simultaneous Ultra-Fast MRI Reconstruction and Tissue Segmentation

**Francesco Calivá**[1]                                                    Francesco.Caliva@ucsf.edu

**Andrew P. Leynes**[2]                                                  Andrew.Leynes@ucsf.edu

**Rutwik Shah**[1]                                                          Rutwik.Shah@ucsf.edu

**Upasana Upadhyay Bharadwaj**[1]                     Upasana.Bharadwaj@ucsf.edu

**Sharmila Majumdar**[1]                                         Sharmila.Majumdar@ucsf.edu

**Peder E. Z. Larson**[1]                                           Peder.Larson@ucsf.edu

**Valentina Pedoia**[1]                                             Valentina.Pedoia@ucsf.edu

[1] *CI², Center for Intelligent Imaging, Department of Radiology and Biomedical Imaging, University of California, San Francisco*
[2] *UC Berkeley – UC San Francisco Joint Graduate Program in Bioengineering*

## Abstract

Magnetic Resonance Image (MRI) acquisition, reconstruction and tissue segmentation are usually considered separate problems. This can be limiting when it comes to rapidly extracting relevant clinical parameters. In many applications, availability of reconstructed images with high fidelity may not be a priority as long as biomarker extraction is reliable and feasible. Built upon this concept, we demonstrate that it is possible to perform tissue segmentation directly from highly undersampled $k$-space and obtain quality results comparable to those in fully-sampled scenarios. We propose '*TB-recon*', a 3D task-based reconstruction framework. *TB-recon* simultaneously reconstructs MRIs from raw data and segments tissues of interest. To do so, we devised a network architecture with a shared encoding path and two task-related decoders where features flow among tasks. We deployed *TB-recon* on a set of (up to 24×) retrospectively undersampled MRIs from the Osteoarthritis Initiative dataset, where we automatically segmented knee cartilage and menisci. An experimental study was conducted showing the superior performance of the proposed method over a combination of a standard MRI reconstruction and segmentation method, as well as alternative deep learning based solutions. In addition, our ablation study highlighted the importance of skip connections among the decoders for the segmentation task. Ultimately, we conducted a reader study, where two musculoskeletal radiologists assessed the proposed model's reconstruction performance.

**Keywords:** fast MRI, task-based MRI reconstruction, multitask deep learning, 3D regression, 3D semantic segmentation, knee cartilage segmentation

## 1. Introduction

Magnetic Resonance Imaging (MRI) enables studying complex tissue structures supported by a remarkable soft tissue contrast. Nevertheless, MRI is not the first imaging technique of choice in many clinical applications. The main reason is its long scanning time, which makes

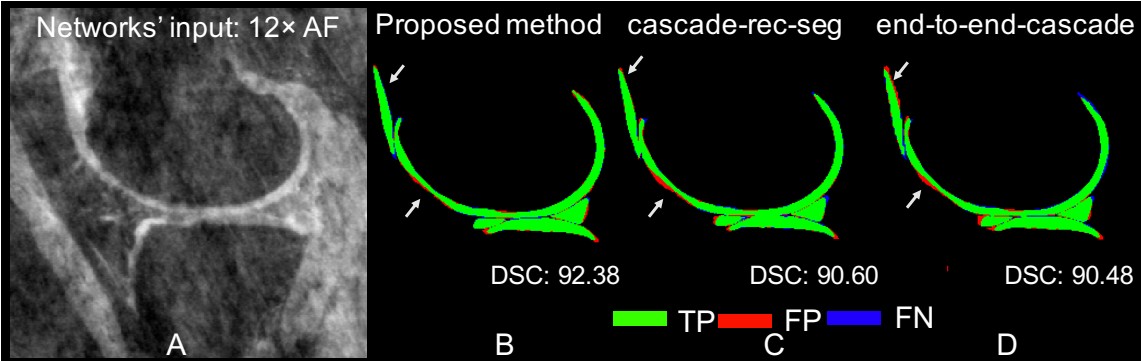

Figure 1: **A)** Example of a 12× undersampled DESS knee MRI that was inputted to the networks. Cartilage and menisci segmentations achieved by **B)** TB-recon, **C)** cascade-rec-seg, **D)** end-to-end-cascade-rec-seg. The reported dice similarity coefficient (DSC) was computed in the shown slice.

it prohibitively slow and expensive (Zbontar et al., 2018; Liang et al., 2019; Chaudhari et al., 2019). Beyond the intrinsic importance of image reconstruction for diagnostic purposes, providing good image quality is crucial for the automatic extraction of clinically valuable biomarkers (Oksuz et al., 2019). To date, fast image acquisition and accurate image post-processing are typically considered separate problems. In this paper, we address accelerated MRI reconstruction and automated tissue segmentation simultaneously. Specifically, we show that by incorporating image reconstruction and segmentation under a unique multitask learning-based framework, one can obtain high quality segmentation at surprisingly high acceleration factors (AFs), as visible in Figure 1. In this paper, segmentation is performed using 4% of the $k$-space acquired data, which – to the best of the authors' knowledge – are 5 times fewer samples than those utilized in Sun et al. (2019), a recent similar work available in the literature.

## 2. Related Work

Task-based reconstruction is a form of multitask learning (*MTL*) (Caruana, 1997). Ruder (2017) illustrates that networks trained on solving more than a single task, outperform networks that were independently trained on each individual task. This happens because although a shared representation is learned for every task, the task-related inductive bias leads the network to learn a more comprehensive and general representation, which all tasks can benefit from (Caruana, 1997). Indirectly, MTL helps improve generalization performance and reduces the risk of overfitting by reducing the model's ability to fit random noise (Bartlett and Mendelson, 2002). We exploit the power of multitask learning by simultaneously solving two tasks, namely reconstruction of undersampled MRIs and multi-class semantic segmentation. A similar problem is addressed in Oksuz et al. (2019), where the correction of motion artefacts in cardiac MRIs is cast to a reconstruction problem, including

simultaneous segmentation of the mid-ventricular tissue. Reconstruction is conducted via a convolutional recurrent neural network, which is followed by a U-Net architecture that solves the segmentation task. This work mainly differs from ours in the fact that no learned representation is shared among tasks. Sun et al. (2019) proposed to utilize a single encoder to perform 3T brain MRI reconstruction (subsequent to the application of a 20% Cartesian under-sampling mask) and brain structure segmentation. Compared to our 3D approach, the framework devised in Sun et al. (2019) consumes cropped 2D patches and reconstructs the MRIs by reproducing CS through a concatenation of 5 U-Nets. In parallel, each of these U-Nets' feature embeddings are decoded to generate segmentation masks, which are ultimately ensembled. Conversely to our work, in Sun et al. (2019), reconstruction and segmentation networks are pretrained independently and fine-tuned under MTL settings. We also achieve noticeably higher acceleration factor, sampling up to only 4% of the $k$-space. Further works regarding accelerated MRI reconstruction and segmentation include but are not limited to Caballero et al. (2014), Schlemper et al. (2018) and Huang et al. (2019b). Apart from being related to different anatomical areas, these studies differ from ours as their image reconstruction pipelines are either iterative Caballero et al. (2014); Huang et al. (2019a,b), or completely by-passed as in Schlemper et al. (2018). Sharing the encoding path among tasks is not a new concept; as a novel addition, in this paper we let features flow among tasks through skip connection between the two decoding paths. We show this ultimately helps improve performance in both tasks. We investigate the performance of our approach in a problem of simultaneous knee MRI reconstruction as well as cartilaginous and meniscal tissues segmentation, solutions to which appear not to be available in the published literature.

## 3. Imaging Dataset and Retrospective Undersampling

The imaging data used in this study are a subset of the Osteoarthritis Initiative (Peterfy et al., 2008), a multi-center longitudinal multi-modality imaging study in 4,796 patients. The selected set comprises 174 3D sagittal double-echo steady-state (DESS) knee MRI scans. They were acquired with a 3.0T Siemens Trio at two time points from 87 patients. Acquisition parameters were TR 16.2ms, TE 4.7ms, FOV 14cm, and readout bandwidth 185kHz, matrix size 384×384×160 and resolution 0.3646×0.3646×0.7mm. For all these volumes, a segmentation of the knee's cartilaginous and meniscal tissues is available with semi-automatic annotations (Peterfy et al., 2008). MRI scans were split into training, validation and test sets, comprised of 119, 28 and 28 volumes respectively, ensuring that patients were not shared across the splits. Prior to under-sampling, MRI data were center-cropped to size 344×344×140, as it was observed this retained relevant structures such as cartilaginous and meniscal tissues. DICOM image data were then reverted to the $k$-space domain by applying a Fourier transform, so that in the $k$-space domain, under-sampling could be performed by applying the undersampling masks. These were generated following a Cartesian retrospective under-sampling approach, which was performed in two directions by means of a variable-density Poisson disk under-sampling mask (Bridson, 2007) achieving 2×, 4×, 6×, 12× and 24× AFs. Subsequent zero-filling and Inverse Fourier Transform

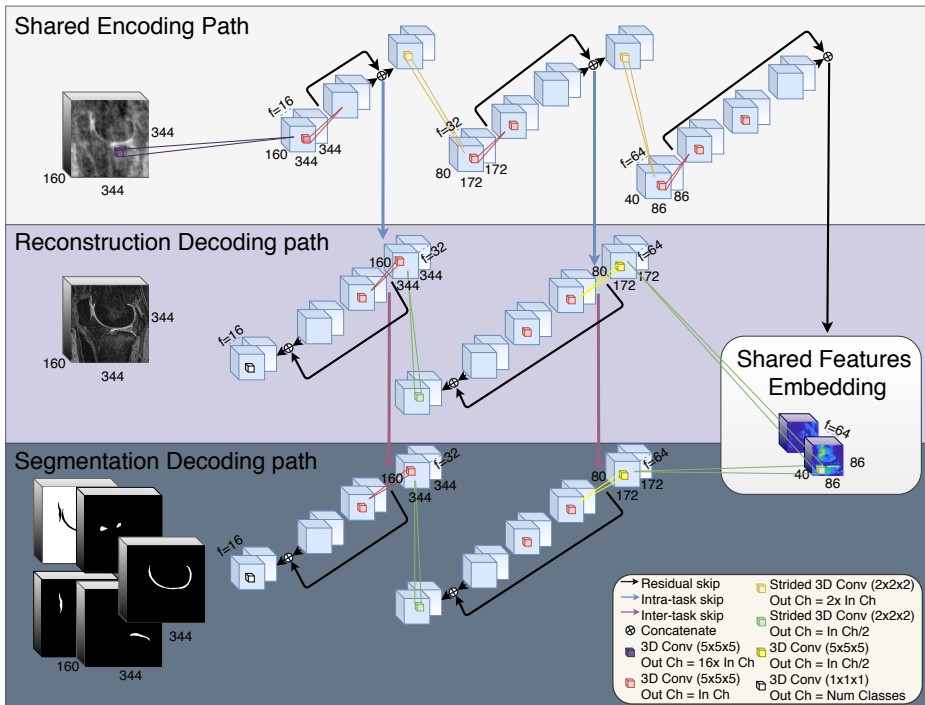

Figure 2: Proposed *TB-recon* architecture. It consumes undersampled MRIs and simultaneously regresses a reconstructed fully-sampled MRI and segments tissues of interest. The architecture suggests a novel way of flowing learned features across multiple tasks (pink arrows). This results in improved performance in both tasks.

completed the generation of undersampled MRIs. The under-sampling was performed using the SigPy software package[1].

## 4. The Proposed Approach

We propose '*TB-recon*', a deep neural network architecture for simultaneous ultra-fast MRI reconstruction and multi-class semantic segmentation. *TB-recon* – which is the short form for 'task-based reconstruction' – is a 3D end-to-end, multitask, fully convolutional encoder-decoder network. To simultaneously solve the tasks of MRI reconstruction from highly undersampled MRIs and multi-class semantic segmentation, our network receives as input a zero-filled $k$-space undersampled knee MRI volume, and produces two outputs: a reconstructed fully-sampled MRI and a multi-class segmentation probability map of cartilaginous and meniscal tissues.

---

1. http://indexsmart.mirasmart.com/ISMRM2019/PDFfiles/4819.html

## 4.1. The Architecture and Implementation Details

The architecture – depicted in Figure 2 – recalls the format of an encoder-decoder system. The encoder is shared among tasks, whereas the decoder includes two task-specific paths. Across the network, in-level flow of learned features is obtained through residual connections (He et al., 2016). This is useful when it comes to reducing the issue of vanishing gradients, through a better flow of gradients during back-propagation. Better features and gradient flows can also be obtained through skip-connection of features between encoding and decoding layers (Ronneberger et al., 2015). We propose a new way of flowing features among the encoding and the two decoding paths, for a more efficient approach to task learning. We use skip-connections between the encoding and the image reconstruction path, as well inter-tasks skip-connections, which provide a direct transfer of features between the two tasks. The reasoning behind such architectural design choice reflects the quality of features passed via skip-connection: at the encoding path, features suffer from under sampled $k$-space artefacts, which especially at high acceleration factors result in a severe loss of finer details that are crucial for tissue segmentation. Along the encoder, the extracted features are instrumental for reconstructing the fully-sampled MRI. In fact, they provide a good initial solution, resulting in a faster convergence to a solution. Arguably, features available along the reconstruction decoder are descriptive of higher quality fine details, fundamental for performing segmentation. The overall number of learned features depends on the number of feature channels that are outputted at the very first encoding convolution, here empirically set to 16. In addition to the task-related features flow, the two decoders differ in their output layers. In the reconstruction side of the network, the output layer is a linear unit and the MRI reconstruction is treated as a regression problem. The segmentation path outputs a 5-channel probability map obtained through a soft-max activation function. Weights are initialized using Xavier method and updated using mini-batch (batch size=1) stochastic gradient descent (SGD), with Adam optimizer. The initial learning rate is 5E-5. The training algorithm minimizes the hybrid loss in Equation (1). This is a linear combination of a reconstruction and segmentation term, where $\alpha$ is a hyper-parameter empirically set to 1.

$$\mathscr{L}_{TB-recon} = \mathscr{L}_{recon} + \alpha \cdot \mathscr{L}_{segm} \tag{1}$$

$\mathscr{L}_{recon}$ – reported in Equation (2) – is a linear combination of a mean absolute error (MAE) and a 3D structural similarity index (SSIM) loss, between the network's logits ($\widehat{y}$) and the fully-sampled target (y).

$$\mathscr{L}_{recon} = 1 - SSIM(\widehat{y}, y) + \beta \cdot MAE(\widehat{y}, y)) \tag{2}$$

The hyper-parameter $\beta$ was empirically set to 6.7 to rescale the two losses (Zhao et al., 2016; Oksuz et al., 2019). A linear combination allows one to better take advantage of the positive aspects of both SSIM and MAE losses (Zhao et al., 2016). SSIM is a measure of similarity between two image patches extracted on a sliding window basis and is highly sensitive to structural information and texture. Since we are reconstructing 3D MRI volumes, SSIM was adapted to handle volumetric data. With respect to the segmentation term, $\mathscr{L}_{segm}$ - reported in Equation (3) - is a multi-class hybrid loss that includes a linear combination of soft dice (Milletari et al., 2016; Sudre et al., 2017) and negative log-likelihood losses.

$$\mathscr{L}_{segm} = 1 - DICE(\widehat{y}, y) + \gamma \cdot NLL(\widehat{y}, y) \tag{3}$$

In Equation (3), $\widehat{y}$ is the predicted semantic segmentation class probability outputted by the soft-max function, y is the target label. Finally, $\gamma$ is a hyper-parameter, empirically set to 0.01 to rescale the two losses.

### 4.2. Comparative Solutions

To compare the effect of multitask learning, including our proposed features flow among tasks, we implemented 2 comparative solutions, namely '*cascade-rec-seg*' and '*end-to-end-cascade-rec-seg*'.

The 3D cascade of reconstruction and segmentation comprises two independent networks, of which the architectures were optimized to perform reconstruction and segmentation respectively. The reconstruction network consumes a 3D undersampled MRI, and returns a fully-sampled MRI. The architecture is a 4 level V-Net-like encoder-decoder (Milletari et al., 2016), where the number of features channels that are outputted at the very first convolution is empirically set to 6. The 3D reconstruction network minimizes a hybrid loss, reported in Equation (2). The semantic segmentation network is trained to segment knee cartilage, including femoral, tibial and patellar cartilage as well as menisci by means of a V-Net-like network, which has the same architecture of the reconstruction network, with the only difference being the output layer, which is a 5 class probability map. The network learns to perform semantic segmentation on fully-sampled DESS MRIs. During training it minimizes a multi-class hybrid loss reported in Equation (3). At test time, the reconstruction network is concatenated downstream with the semantic segmentation network.

The alternative '*end-to-end-cascade-rec-seg*', is a cascade of reconstruction and segmentation networks trained end-to-end. Both sub-networks are V-Net-like architectures having the same number of levels and in-level convolutions as for *TB-recon*. Conversely, the number of feature channels outputted at the very first convolution was set to 8. *End-to-end-cascade-rec-seg* is trained following the same multitask learning settings of *TB-recon*, minimizing a hybrid loss, reported in Equation (1). Here, reconstruction and segmentation tasks have two different encodings, and no features flow among tasks.

## 5. Experimental Study

Multiple sets of experiments were conducted, *i.e.* one for each available acceleration factor; all experiments aiming to MRI reconstruction and cartilaginous as well as meniscal tissue segmentation. All models were trained for 200 epochs using the same train/validation/test splits. To conduct a fair comparison across experiments, we systematically stopped training when no validation improvement was observed for 30 epochs. When training *TB-recon*, *cascade-end-to-end-rec-seg* and the segmentation sub-network of *cascade-rec-seg*, actual validation segmentation dice similarity coefficient (DSC) was monitored. When training the reconstruction sub-network, reconstruction MAE on the validation set was monitored. Training regularization was obtained through Dropout technique with a 95% keep probability and no data augmentation. *TB-recon*'s training required approximately one day per experiment (NVIDIA V100 32GB GPU). At inference, time processing of a volume takes approximately 4s. In contrast, *end-to-end-cascade-rec-seg*'s training required approximately 4 days per experiment and a forward inference pass requires 7s on a machine hosting the same hardware.

With respect to *cascade-rec-seg*, each individual network training required approximately one day per experiment (NVIDIA GTX TITAN X 12GB GPU). At inference time, processing a single volume takes 2s. All implementations are based on Python 3.6.5 and Tensorflow 1.12.0 numerical computation library.

## 5.1. Evaluation

Knee MRI reconstruction performance was quantitatively assessed by means of SSIM and normalized root-mean-square error (NRMSE), and qualitatively by two musculoskeletal (MSK) imaging trained medical doctors. Cartilage and menisci segmentation were assessed by means of DSC. All our metrics were tested for statistical significance ($p \leq 0.05$). We conducted paired $t$-tests to assess whether *TB-recon* significantly outperformed the comparative solutions at all acceleration factors.

Figure 1 compares a segmentation from *TB-recon* and the comparative solutions on a $24\times$ undersampled knee DESS. The arrows point to the trochlea and the articular surface of the patella, showing that *TB-recon* better segmented these particularly challenging areas. Figure 3 is exemplary of a reconstruction from *TB-recon* on a knee DESS undersampled at $6\times$ and $12\times$ AFs. DESS MRI as well as undersampled reconstructed images were inspected by two musculoskeletal imaging trained MDs. On the top and middle rows, bone marrow edema as well as cartilage loss are well preserved after reconstruction at high ($12\times$) AF. On the bottom row, an anterior cruciate ligament (ACL) architecture is completely preserved at $6\times$ AF. A further reader study was conducted, where initially, two MSK MDs underwent a calibration session in which concurrently assessed volumes from 2 subjects in the validation set. Subsequently, both graders independently graded all the reference DESS volumes. MRIs reconstructed by *TB-recon* were distributed such that both readers assessed volumes at a randomized AF order, while blinded to the AF. Contrast, sharpness, SNR and artefacts were the adopted grading metrics. Figure 4 is representative of the reconstruction grading trend with respect to the reference MRIs, which had a baseline grade of 10. Despite a decreasing trend visible at all AFs, 93% of the images $4\times$ undersampled were reconstructed at a reference level quality. Reconstruction quality noticeably decreased at higher AFs: 30% of $12\times$ undersampled were graded of as high quality as the reference DESS. From a more detailed analysis, sharpness and SNR appeared to be the most degraded metrics at higher AFs. Conversely, contrast and lack of artefacts at $12\times$ AF were of reference quality in 71% of the analyzed volumes. At higher acceleration factors, all metrics degraded uniformly irrespective of the high-quality tissue segmentation, which instead was maintained. Experiment results are reported in Figure 5. In the segmentation part, DSC is the average computed on femoral, tibial cartilage and menisci segmentations. These compartments were well segmented also by the comparative solutions, as opposed to the patellar cartilage, for which we report segmentation performance in Appendix in Table 3. Looking at *TB-recon*'s segmentation performance, the DSC differences obtained on $2\times$ and a $24\times$ undersampled MRIs, were less than $1/10$ of the respective DSC standard deviations. Arguably this is a sign that beyond a level of MRI quality, the segmentation is less affected by the reconstruction quality per se. Compared to the alternative solutions, *TB-recon* outperformed them at all acceleration factors. *TB-recon* DSC ranged from $0.8808 \pm 0.0198$ on $2\times$ to $0.8697 \pm 0.0225$ on $12\times$ AF to $0.8563 \pm 0.0256$ on $24\times$ accelerated MRIs, on the

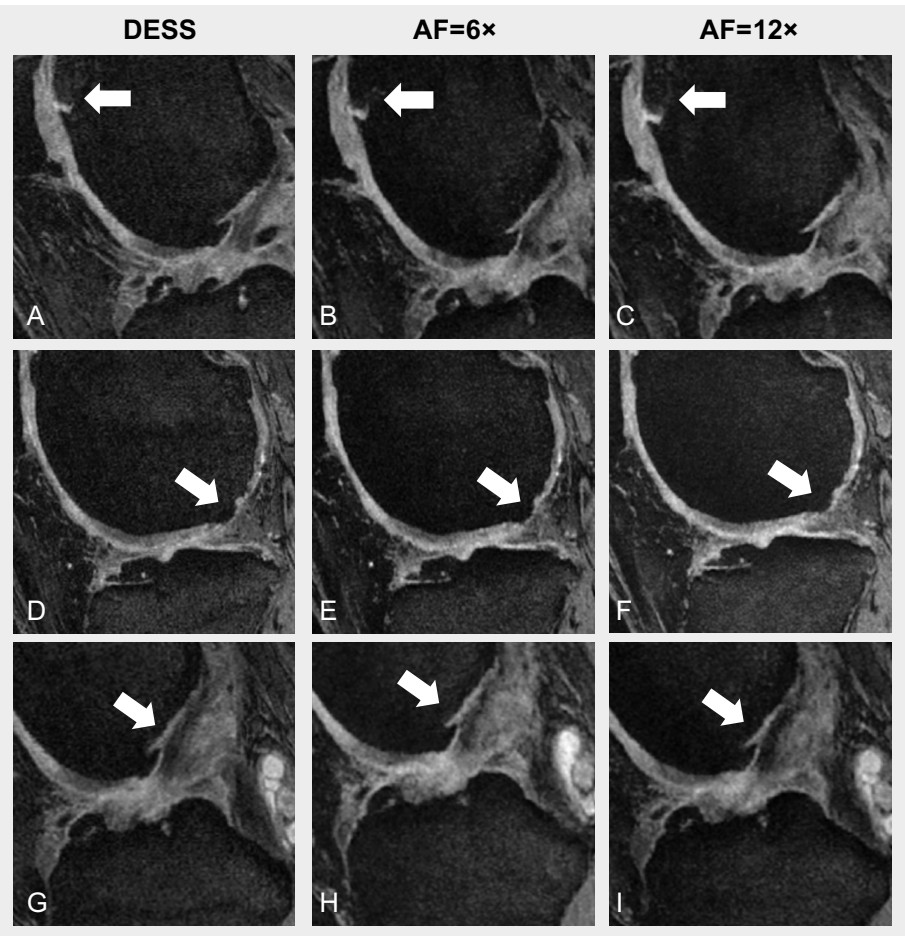

Figure 3: **Top:** bone marrow edema (BME): **A)** sagittal DESS with small area of BME. This finding is well observed in the sequences both at **B)** 6× and **C)** 12× acceleration factors. **Middle:** full thickness cartilage loss: **D)** sagittal DESS with full thickness cartilage loss in lateral femoral cartilage. **E)** same lesion is well observed at 6× and **F)** 12× AFs. **Bottom:** anterior cruciate ligament (ACL): **G)** sagittal DESS with normal ACL architecture. **H)** ACL features are well preserved at 6× AF and partially preserved at **I)** 12× AF.

femoral cartilage. *TB-recon* achieved similar performance on segmenting tibial, patellar cartilage and menisci. Patellar cartilage appeared to be the most challenging compartment, with performance ranging from $0.8217 \pm 0.0609$ (2× AF) to $0.8067 \pm 0.0826$ (12× AF) to $0.7765 \pm 0.0264$ (24× AF). In all 3 compartments segmentation, *TB-recon* significantly outperformed both comparative solutions in processing 24× ultra-fast MRIs (vs *cascade-rec-seg* p=$4.6719e-13$, vs *end-to-end-cascade-rec-seg* p=$8.4980e-05$). On fully-sampled DESS, the segmentation network's reported DSC were $0.8490 \pm 0.0282$, $0.8512 \pm 0.0368$, $0.8139 \pm 0.0698$

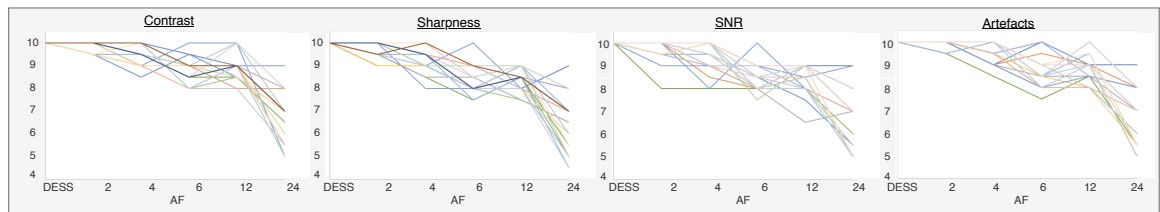

Figure 4: Two musculoskeletal imaging trained MDs assessed TB-recon's reconstruction quality, with respect to the fully-sampled reference DESS MRI. Image assessment involved evaluation of contrast, sharpness, SNR, artefacts and overall image quality. Lines refer to different MRI volumes.

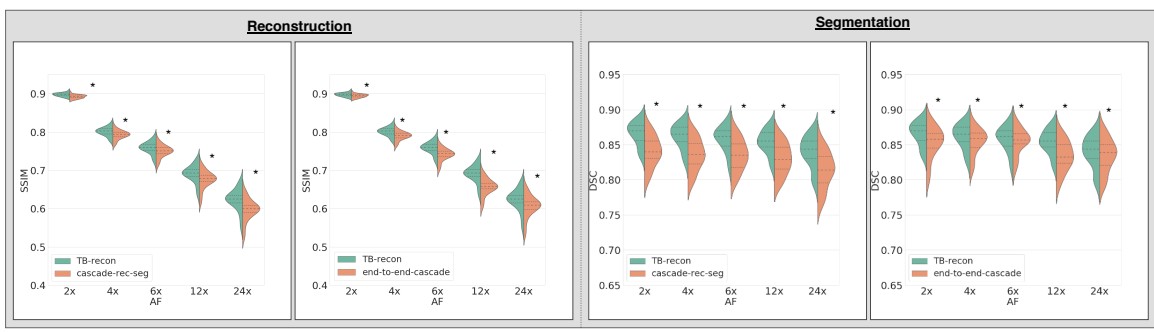

Figure 5: Reconstruction and segmentation performances achieved in the conducted experiments. The star depicts statistical significance (p≤0.05).

in femoral, tibial, patellar cartilage and 0.8361±0.0299 in menisci. *TB-recon*'s significantly better segmentation performance was paired by significant outperformance in reconstructing DESS, at all acceleration factors. This is reported in Figure 5, in terms of SSIM. We report NRMSE in Appendix in Table 4. At the highest available AF, *TB-recon* test performance was SSIM=$0.6929 \pm 0.0229$, NRMSE=$0.2543 \pm 0.0462$. Difference values with respect to the comparative solutions were: for *cascade-rec-seg* $\Delta$SSIM=$0.0260 \pm 0.0021$ (p=$4.8490e - 31$) and $\Delta$NRMSE=$-0.0129 \pm 0.0024$ (p=$1.1350e - 21$). For *end-to-end-cascade-rec-seg* $\Delta$SSIM=$0.0164 \pm 0.0015$ (p=1.7515e-29) and $\Delta$NRMSE=$-0.0133 \pm 0.0020$ (p=$3.1472e - 24$).

## 5.2. Ablation Study and Further Validation

We conducted an ablation study to demonstrate that a) image quality is crucial for high quality segmentation and b) skip connections are key in providing the segmentation task with features representative of higher resolution images. With respect to a), we trained a VNet-like encoder decoder – which essentially is *TB-recon* with a single decoder and a 5

class activated soft-max output – to directly segment zero-filled reconstructed undersampled MRIs with no denoising task involved. We refer to this experimental setup with '*zero-filled*'. With regard to b) we trained a '*naïve*' architecture, which differs from TB-recon in the way skip connections were implemented: from the encoder to each decoder. Table 1 reports the results of this ablation study which highlights the fact that segmentation performance is degraded when segmenting highly undersampled zero-filled data without an intermediate reconstruction step. In addition, we suggest that our introduced skip connections are key in

Table 1: Reconstruction and segmentation performances achieved in the ablation study. *Zero-filled* is an encoder-decoder that directly segments zero-filled reconstructed undersampled MRIs. *Naïve* is an encoder-two-decoders model with encoder-decoders skip connections.

| Femur - DSC | 4× | 6× | 24× |
|---|---|---|---|
| *TB-recon* | **87.33±1.93** | **87.58±1.79** | **85.63±2.56** |
| *zero-filled* | 82.27±2.75 | 81.79±2.73 | 16.27±0.91 |
| *naïve* | 84.97±2.69 | 82.94±2.99 | 83.71±2.82 |

| Tibia - DSC | 4× | 6× | 24× |
|---|---|---|---|
| *TB-recon* | **85.12±3.85** | **86.18±3.59** | **85.84±3.51** |
| *zero-filled* | 83.32±4.53 | 81.26±5.20 | 29.75±1.80 |
| *naïve* | 84.09±3.86 | 83.26±4.20 | 56.15±2.38 |

| Patella - DSC | 4× | 6× | 24× |
|---|---|---|---|
| *TB-recon* | **82.10±6.83** | **81.76±7.82** | **77.65±7.96** |
| *zero-filled* | 78.10±7.17 | 77.37±6.47 | 70.88±10.59 |
| *naïve* | 53.74±4.36 | 77.62±7.61 | 32.45±2.24 |

| Menisci - DSC | 4× | 6× | 24× |
|---|---|---|---|
| *TB-recon* | **84.91±2.79** | **83.78±3.06** | **82.28±2.64** |
| *zero-filled* | 81.87±2.99 | 81.61±3.15 | 16.69±0.93 |
| *naïve* | 82.35±2.69 | 81.47±3.31 | 54.59±2.24 |

the segmentation task; we speculate they allow for a flow of features between reconstruction and segmentation, and these features support segmentation as the reconstruction improves.

Lastly, we investigated how the segmentation only model - used after a traditional reconstruction approach - would perform. Provided that we only considered a single channel acquisition model, in this experiment reconstruction was performed using an L1-Wavelet compressed sensing reconstruction algorithm, as this is suitable to single-coil reconstruction. Compressed sensing (CS) (Lustig et al., 2007) is a promising strategy for fast MRI, where prior to an iterative MRI reconstruction, a reduced number of measurements from a single or multiple receiver channels are sampled below the Nyquist rate. Key drawbacks of CS

involve reduced signal-to-noise-ratio (SNR), as well as overall image quality loss including reduced contrast, sharpness and fine details. The L1-Wavelet approach solves a constrained optimization ($\epsilon = 1E - 6$) problem with a Chambolle-Pock primal-dual hybrid gradient algorithm (Chambolle and Pock, 2011). Subsequent to the CS based reconstruction, we fed the MRIs to the segmentation network. Table 2 shows that the L1-Wavelet CS algorithm provided a very accurate reconstruction at 2x. At higher AFs, the lack of details in the undersampled image negatively affected CS's performance. This is less of a problem for a deep learning method; through a data driven training procedure, priors – which are instrumental for the reconstruction – are introduced. Segmentation results are consistent with those in Table 1. More precisely, a poor reconstruction resulted in a poor segmentation.

Table 2: Reconstruction and segmentation performances achieved in the comparative study against a traditional compressed sensing reconstruction method.

| Average - DSC | 2× | 4× | 6× | 12× | 24× |
|---|---|---|---|---|---|
| *TB-recon* | **85.54±3.58** | **84.87±3.85** | **84.82±4.06** | **84.25±4.28** | **82.25±4.17** |
| L1-Wavelet CS | 82.44±4.89 | 76.3±10.79 | 79.15±5.82 | 67.37±8.99 | 38.94±14.11 |

| SSIM | 2× | 4× | 6× | 12× | 24× |
|---|---|---|---|---|---|
| *TB-recon* | **89.74±0.47** | **80.06±1.16** | **75.84±1.60** | **69.05±2.27** | **62.25±2.8**4 |
| L1-Wavelet CS | 86.9±0.74 | 66.48±2.08 | 65.7±2.10 | 53.11±2.72 | 41.99±2.87 |

| NRMSE | 2× | 4× | 6× | 12× | 24× |
|---|---|---|---|---|---|
| *TB-recon* | **15.80±2.74** | **20.87±3.51** | **22.46±3.77** | **26.06±4.32** | **29.60±4.63** |
| L1-Wavelet CS | 21.80±0.88 | 42.68.±2.54 | 27.83±0.76 | 35.76±1.15 | 49.69±1.25 |

## 6. Discussion

We proposed *TB-recon*, a task-based solution to simultaneously perform segmentation and reconstruction tasks on retrospectively undersampled knee DESS MRIs. *TB-recon* was tested on a wide range of acceleration factors (up to 24×) and demonstrated its capability of producing precise cartilaginous and meniscal 3D segmentation masks in addition to accurate and reliable high resolution reconstructed MRIs. From the experiments, it was observed that a multitask learning strategy improves network performance in both tasks, and that the employment of a shared encoding path as per *TB-recon* results to be more efficient in terms of training time, computational demand and achieved performance. Nevertheless, in this study we did not focus on algorithm's efficiency; the implementation is experimental and could be further optimized. *TB-recon* has the architecture of an encoder-multidecoder system and it leverages the 3D nature of DESS sequences and the multitask learning capability of deep neural networks. An aspect of particular interest in MTL is to assign task importance. MTL can involve joint learning of classification and regression

tasks, at different scales and a naive combination of the task-specific losses might not always be the best solution. We gave equal importance to both tasks, even if the actual goal was to obtain a precise segmentation from ultra-fast MRIs. Oksuz et al. (2019) showed that task weighting can highly impact performance on the tasks on hand. We leave a systematic assessment of such impact as part of our future work. Based on the results obtained in our experimental study, we speculate that in a dynamic weighting setting, both tasks would be weighted in a way such that, at the initial training stage reconstruction was the main focus. As reconstructed image quality improves, the segmentation task should be weighted more, leading to the desired segmentation.

While these results are promising there are some limitation to be acknowledged. Volume MRIs were retrospectively undersampled from DICOM files. The procedure of reverting a DICOM image to the $k$-space domain by applying a Fourier transform does not lead to the originally measured MRI raw data (Zbontar et al., 2018; Hammernik and Knoll, 2020). This is mainly due to the fact that DICOM files are usually the output of acquisition and post-processing algorithms, which causes a discrepancy between our synthesized $k$-space data and the actual acquired $k$-space data. This discrepancy includes the loss of image phase information, crucial for the image generation. Nonetheless, we expect that the results we obtained would translate to an actual acquisition with true acquired data. Furthermore, starting from DICOM files indirectly forced us to treat the problem as single-coil MRI reconstruction. Extension from single-coil to a multi-coil reconstruction is not straightforward and is an open research question (Souza et al., 2019), especially because images from different coils carry complementary information. Traditional methods for multi-channel acquisitions involve parallel imaging reconstruction methods, such as GRAPPA (Griswold et al., 2002) or SENSE (Pruessmann et al., 1999). Theoretically, deep learning-based approaches should outperform GRAPPA and SENSE on multi-channel data, because GRAPPA and SENSE methods could be treated as linear convolutional layers. Deep learning-based approaches for image reconstruction take advantage of multiple layers and non-linearity to learn weighting and even improve performance by overcoming model's imperfections. This was further confirmed by Schlemper et al. (2019)'s performance in the NeurIPS 2019 Fast MRI challenge (Knoll et al., 2020). Furthermore, iterative reconstruction algorithms like CS are inherently complex and often require long reconstruction times, making their deployment challenging in daily clinical practice (Hammernik and Knoll, 2020). Another limitation to consider in this work is that data consistency operations were not incorporated in the network architecture such as those found in unrolled iterative optimization networks (Wu et al., 2019; Diamond et al., 2017). This was not possible due to lack of ground-truth raw data.

## 7. Conclusions

We proposed *TB-recon*, a solution for simultaneous reconstruction and segmentation to enable ultra-fast MRI. In this retrospective study, *TB-recon* demonstrated the data-driven nature of DL-based solutions has a potential to make ultra-fast MRI feasible. We argue that task-based reconstruction can push the boundaries of fast MRI far beyond the acceleration factors that have been utilized in previous works. With the conducted experiments we demonstrated this is not just an incremental improvement. By combining the image reconstruction with an image interpretation task, we forecast to break previous speed limits,

which have hampered the application of magnetic resonance imaging. We strongly believe that MRI practice can benefit from task-based reconstruction, with potential application to well defined tasks. Applications of this may be disease and abnormality identification, segmentation of the gray/ white matter and other structures in the brain, estimation of the volume of organs, size of various structures including cartilage thickness and lesion size and counting for pathologies such as multiple sclerosis and micro-bleeds. We hope this paper further stimulates the research community's interest on task-based fast MRI.

## Acknowledgments

This work was supported by the NIH/NIAMS R00AR070902 grant. We would like to thank Adam Noworoloski, Xucheng Zhu, Claudia Iriondo and Kaiyang Cheng for the help with the project and fruitful discussions. We would also like to thank Miguel Monteiro for the VNet implementation available at https://github.com/MiguelMonteiro/VNet-Tensorflow. Ultimately, we would like to thank the reviewers for their constructive feedback and their efforts towards improving our manuscript.

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

## Appendix A. Further Experimental Results

In addition to 5.1, we report segmentation performance in the patella, and the reconstruction NRMSE.

Table 3: Patellar cartilage segmentation quality in DSC metric. Methods performance comparison at all acceleration factors. Patellar cartilage segmentation performance achieved by the vanilla segmentation network on fully-sampled DESS was 0.8139±0.0698.

| AF | TB-recon | cascade-rec-seg | end-to-end-cascade |
|-----|-----------|-----------------|---------------------|
| 2× | **82.17±6.09** | 80.75±7.02 | 53.54±6.01 |
| 4× | **82.10±6.83** | 80.46±7.18 | 54.40±5.61 |
| 6× | **81.76±7.82** | 79.43±6.95 | 55.20±4.58 |
| 12× | **80.67±8.26** | 79.06±7.73 | 53.13±5.06 |
| 24× | **77.65±7.96** | 76.80±8.49 | 53.45±5.34 |

Table 4: Reconstruction quality in NRMSE metric. Methods performance comparison at all acceleration factors.

| AF | TB-recon | cascade-rec-seg | end-to-end-cascade |
|-----|-----------|-----------------|---------------------|
| 2× | **15.80±2.74** | 16.24±0.51 | 16.55±2.92 |
| 4× | **20.87±3.51** | 21.65±0.75 | 22.60±3.90 |
| 6× | **22.46±3.77** | 23.05±0.86 | 25.42±4.40 |
| 12× | **26.06±4.32** | 26.76±1.02 | 34.55±6.01 |
| 24× | **29.69±4.93** | 30.97±1.20 | 31.02±5.17 |

## Appendix B. Experiment With Standardized Architecture Parameters

Comparative solutions were designed to maximize performance on the specific tasks each networks solved. Various solutions were investigated, and the best performing models were selected to be the comparative solutions. We experienced that having a larger number of filters at the input layers does not necessarily reflect better performance in the task; similarly we could not define a relationship between number of network levels and segmentation performance. With that in mind, we conducted an additional experiment with standardized networks, using the undersampled dataset at 24× AF. In practice, in this experiment, all architectures had 8 feature channels at the input layer, 2 levels with 1 and 2 convolutions per-level respectively, and 3 additional convolutions at the bottom level. Table 5 reports the results of this experiment. On average, results show that TB-recon segments the tissues of interested better than the comparative solutions, since it was able to extract a higher

quality segmentation of the patella. The reconstruction quality among the three methods were comparable with no significant difference. We speculate – in agreement with the results reported in Table 1 – that the skip connections between the tasks are key for the improved segmentation. In practice, the metrics utilized to assess image quality merely provide an average estimate of the reconstruction quality, which does not reflect whether tissues of interest were reconstructed with fidelity (for instance, *cascade-rec-seg* is particularly exemplary of this). From a visual inspection, we observed that our proposed network better reconstructed the regions where the tissues of interest are positioned. We hypothesize this is also due to the presence of inter-task skip connections, which could encourage collaboration between the two tasks. Nonetheless, further investigating image quality within the segmentation region is of great interest and part of our future research.

Table 5: Reconstruction and segmentation performances achieved in the experiment with standardized architecture parameters. The experiment is conducted on retrospectively undersampled MRIs at 24× AF.

| DSC | Average | Femur | Tibia | Patella | Menisci |
|---|---|---|---|---|---|
| *TB-recon*-8ch | **81.55±0.58** | 84.19±2.95 | 82.95±4.66 | **78.15±7.70** | 80.94±2.64 |
| *cascade-ETE*-8ch | 75.92±2.15 | 84.75±2.70 | 83.79±3.80 | 53.46±5.34 | 81.67±2.72 |
| *cascade-rec-seg*-8ch | 44.86±29.43 | 60.46±29.22 | 58.47±31.36 | 0.0 | 60.49±28.3 |

| Input: 24× AF | SSIM | NRMSE |
|---|---|---|
| *TB-recon*-8ch | 59.95±2.74 | 31.49±5.29 |
| *cascade-ETE*-8ch | 60.61±2.82 | 31.02±5.17 |
| *cascade-rec-seg*-8ch | 60.40±2.83 | 30.46±1.19 |

