# OpenReview forum: "Breaking Speed Limits with Simultaneous Ultra-Fast MRI Reconstruction and Tissue Segmentation"
_MIDL.io/2020/Conference — MIDL 2020_

### Official Review · AnonReviewer1 · 2020-03-11
**A CNN-based method is introduced for joint MRI reconstruction and segmentation using undersampled K-space data.**

**Rating:** 4
**Confidence:** 5
**Recommendation:** Oral

**Summary:**

Standard imaging analysis pipeline does imaging reconstruction and segmentation in a sequential way. As a result, high-quality image are usually needed to improve segmentation results. This paper introduced a novel framework for simultaneously image reconstruction and segmentation using a CNN-based method.

**Strengths:**

The CNN-based method imposed the same encoding path for the image reconstruction and segmentation tasks.
The image reconstructions and segmentation path has the shared feature embeding. Moreover, the network was trained using an objective function that combine the reconstruction loss and segmentation loss. Experimental results show that this method reduces false positive and false negative segmentations than the cascade-rec-seg and end-to-end-cascade methods. The weakness of this work was also reasonably adressed.

**Weaknesses:**

I don't see obvious weakness of this work besides the comments mentioned by the authors in the paper. I suggest that the authors can revise Figure 3 to zoom into the details so that they can be seen more easily.

**Justification Of Rating:**

It is a very reasonable application for a form of multitask learning (MTL) to solve the problem of simultaneous image reconstruction and segmentation. This method has better performance than the cascade analysis methods.

**Paper Type:**

methodological development

**Questions To Address In The Rebuttal:**

If possible, Figure 5 can be improved so that the three methods can be displayed in the same figure.

**Special Issue:**

yes

---

> ### Author Response · Authors · 2020-03-25
> **Response to Reviewer 1**
>
> We thank the reviewer for the interest in our work and the positive feedback they provided.
>
> Q1: Revise Figure 3 to zoom into the details so that they can be seen more easily. If possible, Figure 5 can be improved so that the three methods can be displayed in the same figure.
>
> A1: We appreciate the request to improve the quality of Figures 3 and 5. As a result, we will update these figures accordingly in the revised manuscript.

---

### Official Review · AnonReviewer4 · 2020-03-13
**Comparative study is questionable**

**Rating:** 2
**Confidence:** 4
**Recommendation:** Poster

**Summary:**

The authors used a multitask learning approach for simultaneous reconstruction and segmentation on structural knee MRI. The proposed model was compared to two cascaded reconstruction and segmentation networks. The results suggest that the multitask approach improved both reconstruction and segmentation tasks.

**Strengths:**

Unlike general multitask approaches where the tasks are separated in late layers, the authors designed the CNN architecture sharing encoding path but separate decoding paths with skip connections which is the strength of the paper.

**Weaknesses:**

Comparative studies need to be performed better as in its current form they do not provide a clear indication of why the proposed method provides better results. One of the main design criteria to consider is the change in the number of initial feature channels and layers. The number feature channels are 16 on the proposed network, 6 in cascade-rec-seg and 8 in end-to-end-cascade-rec-seg. The reason was identified as the GPU memory limitations but it is not clear why it was not kept e.g. as 6 for all the networks. Moreover, 2 level encoder-decoder is used in the proposed method but it was changed to a 4-level version in other models. These changes in the comparative study make the results questionable.
Since both recon and segmentation is learned in this study, it is not clear how the segmentation only model  - used after a traditional reconstruction using grappa or sense type of approaches - would compare to the results presented in this work.

**Detailed Comments:**

It is not clear why inter-task skip connections were employed.  Please analyze the effect of inter-task skip connections in the accuracy of segmentation and reconstruction.
The following sentence in the discussion is an overstatement, consider deleting it "This opens new horizons in MRI practice, with potential applications including dynamic MRI, automated fast and personalized MRI protocol modications, currently unfeasible with the state-of-the-art technology."




**Justification Of Rating:**

There are so many changes in the architectures for the comparitive study which would directly affect the results presented in this work. These types of changes need to be minimized to make the results convincing.

**Paper Type:**

methodological development

**Special Issue:**

no

---

> ### Author Response · Authors · 2020-03-26
> **Response to Reviewer 4 part 1**
>
> We appreciate the reviewer’s enthusiasm towards using a shared encoder with skip connection between the decoders, and their advice to improve the quality of our paper.
>
> Q1: Comparative studies need to be performed better… to consider is the change in the number of initial feature channels and layers… it is not clear why it was not kept e.g. as 6 for all the networks.
>
> A1: We understand that our word choice gave rise to suspicions about the performance of the comparative solutions, which we presented. We believe it is necessary to rephrase our justification for having comparative solutions with different architectures. Our comparative solutions were designed so that the task performance each network solves is maximized. Our comment about maximizing available GPU memory is misleading.
> We investigated many solutions to keep comparisons as fair as possible; based on performance on the validation set, we then selected the networks that best performed and utilized them as comparative solutions. In other words, we wanted to compare TB-recon with top performing networks. We experienced that having a larger number of filters at the input layers does not always reflect better performance in the task; similarly we cannot define a relationship between number of network levels and segmentation performance.
> With that in mind, we are currently conducting an experiment with standardized networks:
> -8 feature channels
> -2 levels
> -1,2 convolutions per level
> -3 bottom convolutions
> This choice did not require re-training cascade-ETE. We will report the results of this experiment when complete.
>
> Q2: it is not clear how the segmentation only model - used after a traditional reconstruction using grappa or sense type of approaches - would compare.
> A2: In this work, we only considered a single channel acquisition model. For this reason, in response to your request to compare our method with a traditional reconstruction, we compared the proposed method with a compressed sensing reconstruction. We employed an L1-Wavelet compressed sensing [1,2] reconstruction algorithm which is suitable to single-coil reconstruction. Subsequent to the CS based reconstruction, we fed the MRIs to the segmentation network.
>
> Metric: Average DSC
> AF |Proposed  |L1-Wavelet CS
> 2x  |85.54±3.58|82.44±4.89
> 4x  |84.87±3.85|76.3±10.79
> 6x  |84.82±4.06|79.15 ± 5.82
> 12x|84.25±4.28|67.37 ± 8.99
> 24x|82.25±4.17|38.94 ±14.11
>
> Metric SSIM
> AF | Proposed  |L1-Wav CS
> 2x  |89.74±0.47 |86.9±0.74
> 4x  |80.06±1.16 |66.48±2.08
> 6x  |75.84±1.60 |65.7±2.10
> 12x|69.05±2.27 |53.11±2.72
> 24x|62.25±2.84 |41.99±2.87
>
> Metric NRMSE
> AF |Proposed  |L1-Wav CS
> 2x  |15.80±2.74|21.80±0.88
> 4x  |20.87±3.51|42.68.±2.54
> 6x  |22.46±3.77|27.83±0.76
> 12x|26.06±4.32|35.76±1.15
> 24x|29.60±4.63|49.69±1.25
>
> We agree we could use a multi-channel acquisition model, and a reconstruction parallel imaging method, such as GRAPPA or SENSE. Theoretically, DL approaches should outperform GRAPPA and SENSE on multi-channel data, because GRAPPA and SENSE methods could be treated as fixed weighting convolution layers without non-linear activations. DL approaches for image reconstruction take advantage of multiple layers and non-linearity to learn weighting and even improve performance by overcoming model’s imperfections.
>
> Q3: Comparative study is questionable.
> A3: To further validate our results, for each AF, we trained a segmentation net (same of cascade-rec-seg) augmenting the training set by using MRIs that were reconstructed using the recon network. The reason for this experiment is that since TB-recon is trained end-to-end from undersampled images, it also utilizes features that are representative of reconstructed images. Conversely, the segmentation network of cascade-rec-seg was trained on fully sampled images but tested on reconstructed images. With this experiment we show that the test sample distribution shift for the segmentation network cannot explain its performance drop compared to TB-recon.  From the experiment, the non-augmented networks performed significantly better (p$\leq$0.05) than the augmented counterparts at all AFs, in all tissues. We can only speculate motivations for this result; it may be due to the fact that the reconstruction network provides smoother images, where objects’ boundaries lose sharpness and it is harder to train a segmentation network for cartilage segmentation, as this is a very limited structure and its segmentation highly depends on the correct edge detection.
>
> Q4: Please analyze the effect of inter-task skip connections.
> A4: We are currently training a 'naïve' MTL architecture (shared encoder with encoder-decoders skip connections) and we will report the results of this experiment when complete.
>
> [1] M. Lustig, et al. “Compressed sensing MRI,” IEEE Signal Process. Mag., vol. 25, no. 2, p. 72, 2008.
> [2] http://indexsmart.mirasmart.com/ISMRM2019/PDFfiles/4819.html

---

> > ### Author Response · Authors · 2020-03-28
> > **Response to Reviewer 4 part 2**
> >
> > A4: We conducted two experiments which show that a) image quality is crucial for high quality segmentations and b) skip connections are key in introducing to the segmentation task features which are representative of higher resolution images, and yet these features’ quality enhances as the reconstruction improves.
> >
> > With regard to a), we trained a VNet-like encoder decoder – which like TB-recon is comprised of 2 levels with 1 and 2 conv per level, 3 conv at the bottom level and 16 feature channels at the input layer. The network is trained to only segment zero-filled reconstructed MRIs with no denoising task involved and we refer to it with ‘zero-filled’. With respect to b) we trained a 'naïve' architecture, which differs from TB-recon in the way skip connections are implemented, as per the comment received. We report the results obtained at 4x, 6x and 24x. Since the remaining 2 networks have not completed training at the rebuttal submission deadline, we will report results in the amended manuscript.
> >
> > Femur - DSC
> > Method               4x        	6x          	    24x
> > Proposed      	87.33±1.93  87.58±1.79   85.63±2.56
> > zero-filled   	82.27±2.75  81.79±2.73   16.27+0.91
> > Naive             	84.97±2.69  82.94±2.99    83.71±2.82
> >
> > Tibia - DSC
> > Method                  4x         		6x          	  24x
> > Proposed       85.12±3.85	86.18±3.59	85.84±3.51
> > zero-filled   	83.32±4.53  	81.26±5.20  	29.75±1.80
> > Naive              84.09±3.86  	83.26±4.20  	56.15±2.38
> >
> > Patella - DSC
> > Method                   4x         	 6x          	        24x
> > Proposed       82.10±6.83  81.76±7.82   77.65±7.96
> > zero-filled   	78.10±7.17  77.37±6.47    70.88±10.59
> > Naive              53.74±4.36  77.62±7.61    32.45±2.24
> >
> > Menisci - DSC
> > Method                    4x               6x                    24x
> > Proposed       84.91±2.79	83.78±3.06	82.28±2.64
> > zero-filled   	81.87±2.99  	81.61±3.15    	16.69±0.93
> > Naive              82.35±2.69  	81.47±3.31    	54.59±2.24
> >
> > We argue that the results in these experiments show that by incorporating a reconstruction step to the segmentation procedure, segmentation quality is enhanced, especially at high acceleration factors where in most of the compartments poor segmentations were obtained with ‘zero-filled’. This is less of a problem for the other methods, since we observed a noticeable improvement which we attribute mainly to the image reconstruction task involved. In addition, we suggest that this experiment shows that our introduced skip connections are key in the segmentation task; we speculate they allow for a flow of features between reconstruction and segmentation, and these features support segmentation as the reconstruction improves.
> >
> > A1: In addition, to our previous answer to Q1, as anticipated we conducted an experiment with standardized networks, using data retrospectively accelerated at 24x AF.
> >
> > DSC (input 24xAF)
> > Method        	                   Avg              Femur       	Tibia     	Patella           Menisci
> > proposed 8ch               81.55±0.58   84.19±2.95    82.95±4.66  78.15±7.70 80.94±2.64
> > cascade-ETE                  75.92+2.15   84.75±2.70    83.79±3.80  53.46±5.34 81.67±2.72
> > cascade-rec-seg 8ch   44.86+29.43  60.46±29.22  58.47±31.36     0.0          60.49±28.3
> >
> > Method        	                  SSIM	        NRMSE
> > proposed 8ch             59.95+2.74   31.49±5.29
> > cascade-ETE                60.61±2.82  31.02±5.17
> > cascade-rec-seg 8ch  60.40±2.83   30.46±1.19
> >
> > From the results we observed that on average our proposed method can produce a better segmentation than the comparative solutions, since it was able to extract a higher quality segmentation of the patella. The reconstruction quality among the three methods were comparable with no significant difference. We speculate - in agreement with the results of the experiment reported in A4 - that the skip connections between the tasks are key for the improved segmentation. In practice, the metrics utilized to assess image quality merely provide an average estimate of the reconstruction quality, which does not reflect whether tissues of interest were reconstructed with fidelity (for instance, cascade-rec-seg is particularly exemplary of this). From a visual inspection, we observed that our proposed network better reconstructed the regions where the tissues of interest are positioned. We hypothesize this is also due to the presence of inter-task skip connections, which could encourage collaboration between the two tasks. Nonetheless, further investigating image quality within the segmentation region is of great interest in the future.

---

### Official Review · AnonReviewer2 · 2020-03-15
**This is an interesting paper despite some weaknesses. I would recommend rebuttle.**

**Rating:** 3
**Confidence:** 3
**Recommendation:** Poster

**Summary:**

The authors proposed a TB-recon network to perform image reconstruction and tissue segmentation simultaneously from undersampled DESS knee MRI rather than treating them as separate problems. The unique aspects of the proposed network are (1) sharing the encoding path (which is not new) and (2) introducing intra-task skip connections in the decoding path allowing information flow between tasks. The authors compared the proposed network by solving the two tasks sequentially independently (cascade-rec-seg) and jointly (end-to-end cascade-rec-seg). Experiment results showed that the proposed network significantly outperforms the other two alternatives.

**Strengths:**

The problem is interesting, and the authors proposed a comprehensive technique to tackle it.

One important finding of this paper is that by performing image segmentation and image reconstruction at the same time actually improves the quality of both tasks compared to performing them independently.

It is nice to musculoskeletal imaging trained MDs to evaluate the actual quality of the image rather than just depend on evaluation metrics.


**Weaknesses:**

Having the intra-task skip connections is an important novel aspect of the pipeline. However, there is no experiment demonstrates the importance of these connections. It is possible that the network will perform well without these skip connections.

The authors us the same set of hyper-parameters (weights between losses) optimized for the proposed work in the end-to-end-cascade-rec-seg experiment, which may not be optimal to the latter approach. Indeed, the image reconstruction and segmentation quality are much worse even compared to the cascade-rec-seg experiment. The authors claimed that its sub-optimal performance is because it is much more computational demanding, which I have trouble understanding.

Lack of comparisons with other state-of-the-art approaches.


**Detailed Comments:**

Typo in the abstract: “it that is” should be “that it is”

It would be nice to have the Dice scores of the three methods in Figure 1.


**Justification Of Rating:**

Although there are some weaknesses, overall, the paper is well written and well organized. The problem is interesting, and the results are promising. If the authors can address the comments sufficiently, I would recommend the acceptance of this paper.

**Paper Type:**

methodological development

**Questions To Address In The Rebuttal:**

Perform experiment to demonstrate the importance of intra-task skip connections.

Get the best performance of the end-to-end-cascade-rec-seg experiment or explain why it is much more expensive to perform.


**Special Issue:**

yes

---

> ### Author Response · Authors · 2020-03-26
> **Response to Reviewer 2 part 1**
>
> We thank the reviewer for the constructive feedback they provided, with suggestions to improve the manuscript.
>
> Q1: Perform experiment to demonstrate the importance of intra-task skip connections.
> A1: We appreciate your curiosity about the importance of sharing representation among tasks, which is obtained through skip connections between the two decoders as opposed to traditional encoder-decoder skip connections. We are currently training a 'naïve' MTL architecture (shared encoder with encoder-decoders skip connections) and we will report the results of this experiment when complete.
>
> Q2: Get the best performance of the end-to-end-cascade-rec-seg experiment or explain why it is much more expensive to perform.
> A2: According to multitask (MT) theory, when tasks are correlated, a network trained under MT settings performs better than many networks that are each trained for a task. We, along with others [1], show that reconstruction and segmentation are tasks suitable for MTL. Based on this, we expect training the cascade-rec-seg in end-to-end settings to improve performance in both tasks. Memory constraints did not allow such combination (not even to use a TB-recon-like architecture with its same num per-level convolutions and 16feature channels). We acknowledge this as a limitation and believe with multi-GPUs training it might be possible to test bigger networks. Nonetheless, we utilize the best performing ETE architecture available in our experiments.
>
> Q3: Lack of comparisons with other state-of-the-art approaches.
> A3: We compared our segmentation results with the top 5 entries in the cartilage and menisci segmentation challenge of IWOAI 2019 [3], since the test set is the same. TB-recon is competitive even when the inputted MRI is retrospectively 12x under-sampled. Deniz et al. [7] implemented a 3D U-Net with dilated convolutions and minimized a hybrid loss including DICE and cross-entropy. Gaj et al. [4] added dense convolutions to DeeplabV3+. Iriondo et al. proposed an ensemble of 2D and 3D VNets. Perslev et al. [6] fed 2D slices from multiple planes to a U-Net. Khosravan et al. [5] employed a GAN based segmentation framework.
> Team	      	 | Femur	| Tibia		| Patella 	| Menisci	| Average
> Gaj et al.   	 |90.49±1.59	|88.76±3.51	|85.62±7.14	|87.38±3.28	|88.06±3.88
> Iriondo et al	 |89.98±1.88	|88.58±3.24	|85.76±6.39	|87.76±2.79	|88.02±3.57
> Perslev et al.	 |90.11±1.61	|88.83±2.61	|84.91±9.77	|87.75 ±2.46	|87.90±4.11
> Deniz et al 	 |87.79±2.31	|87.20±3.36	|82.68 ±8.22	|83.93±2.83	|85.40±4.18
> Proposed 12x |86.97±2.25	|85.84±3.51	|80.67±8.26	|83.29±3.14	|84.19±4.29
> Khosravan et al.|86.68±2.62|84.90±4.29	|81.16±9.23	|83.49±3.14	|84.06±4.82
>
> Q4: The authors use the same set of hyper-parameters (weights between losses) optimized for the proposed work in the end-to-end-cascade-rec-seg experiment, which may not be optimal to the latter approach.
> A4: We thank the reviewer for the comment. We acknowledged this as a limitation in the discussion session. Setting $\alpha=1$ in Eq. 1 results in giving equal importance to both tasks. This was not optimized for our approach, and we committed to a systematic assessment of tuning alpha in our future work. We believe that dynamic weighting of the loss terms is the direction to go. A possible solution is proposed by [2]. We imagine that in dynamic weighting settings, the network will first focus on the reconstruction task, bringing more attention to the segmentation task at a later stage, as such behavior would be coherent with our presented experiments (including those in the rebuttal) showing that good quality reconstruction is necessary (but not sufficient) for good quality segmentation.
>
> Q5: It would be nice to have the Dice scores of the three methods in Figure 1.
> A5: We will amend Figure 1 in the revised version of the manuscript.
>
> [1] Oksuz, Ilkay, et al. "Deep Learning Based Detection and Correction of Cardiac MR Motion Artefacts During Reconstruction for High-Quality Segmentation." arXiv preprint arXiv:1910.05370 (2019).
> [2] Kendall, Alex, et al. "Multi-task learning using uncertainty to weigh losses for scene geometry and semantics." Proceedings of the IEEE conference on computer vision and pattern recognition. 2018.
> [3]https://www.isoai.org/annualconferenceiwoai
> [4]Gaj S, et al. "Automated cartilage and meniscus segmentation of knee MRI with conditional generative adversarial networks". Magnetic Resonance in Medicine
> [5]Khosravan N, et al., "PAN: Projective Adversarial Network for Medical Image Segmentation."  arXiv preprint arXiv:1906.04378
> [6]Perslev M, et al., "One Network to Segment Them All: A General, Lightweight System for Accurate 3D Medical Image Segmentation." http://link.springer.com/10.1007/978-3-030-32245-8_4
> [7]Deniz CM, et al.,"Segmentation of the Proximal Femur from MR Images using Deep Convolutional Neural Networks". Sci Rep. 2018.

---

> > ### Author Response · Authors · 2020-03-28
> > **Response to Reviewer 2 part 2**
> >
> > A1: We conducted two experiments which show that a) image quality is crucial for high quality segmentations and b) skip connections are key in introducing to the segmentation task features which are representative of higher resolution images, and yet these features’ quality enhances as the reconstruction improves.
> >
> > With regard to a), we trained a VNet-like encoder decoder – which like TB-recon is comprised of 2 levels with 1 and 2 conv per level, 3 conv at the bottom level and 16 feature channels at the input layer. The network is trained to only segment zero-filled reconstructed MRIs with no denoising task involved and we refer to it with ‘zero-filled’. With respect to b) we trained a ‘naïve’ architecture, which differs from TB-recon in the way skip connections are implemented, as per the comment received. We report the results obtained at 4x, 6x and 24x. Since the remaining 2 networks have not completed training at the rebuttal submission deadline, we will report results in the amended manuscript.
> >
> > Femur - DSC
> > Method                  4x        	6x          	    24x
> > Proposed       87.33±1.93  87.58±1.79  85.63±2.56
> > zero-filled   	82.27±2.75  81.79±2.73  16.27+0.91
> > Naive              84.97±2.69  82.94±2.99  83.71±2.82
> >
> > Tibia - DSC
> > Method                   4x         		6x          	24x
> > Proposed       85.12±3.85	86.18±3.59	85.84±3.51
> > zero-filled   	83.32±4.53  	81.26±5.20  	29.75±1.80
> > Naive             	84.09±3.86  	83.26±4.20  	56.15±2.38
> >
> > Patella - DSC
> > Method                   4x         	 6x          	      24x
> > Proposed       82.10±6.83  81.76±7.82   77.65±7.96
> > zero-filled   	78.10±7.17  77.37±6.47    70.88±10.59
> > Naive              53.74±4.36  77.62±7.61    32.45±2.24
> >
> > Menisci - DSC
> > Method                    4x                6x                   24x
> > Proposed       84.91±2.79	83.78±3.06	82.28±2.64
> > zero-filled   	81.87±2.99  	81.61±3.15    	16.69±0.93
> > Naive              82.35±2.69  	81.47±3.31    	54.59±2.24
> >
> > We argue that the results in these experiments show that by incorporating a reconstruction step to the segmentation procedure, segmentation quality is enhanced, especially at high acceleration factors where in most of the compartments poor segmentations were obtained with ‘zero-filled’. This is less of a problem for the other methods, since we observed a noticeable improvement which we attribute mainly to the image reconstruction task involved. In addition, we suggest that this experiment shows that our introduced skip connections are key in the segmentation task, since they allow for a flow of features between reconstruction and segmentation, and these features enhance segmentation as the reconstruction improves.

---

### Official Review · AnonReviewer3 · 2020-03-20
**Multi-task learning framework for reconstruction + segmentation from under-sampled MR data. More comparisons needed.**

**Rating:** 2
**Confidence:** 3

**Summary:**

The paper proposes a multi-task learning framework for MRI reconstruction and segmentation from under-sampled k-space data. Results indicate that fairly accurate segmentations can be obtained already with highly under-sampled data. The proposed method is compared with two variants, which indicate (1) end-to-end learning of both tasks and (2) sharing encoder features is beneficial for segmentation performance.

**Strengths:**

* The paper tackles an important, but relatively under-studied problem of doing segmentation from under-sampled k-space data. Such a method could have high potential in accelerating personalized MR sequencing.

* For the methods which are compared against, the experiments are carried out in a structured manner and the results well-presented.

**Weaknesses:**

* The main methodological novelty of the paper is a change in the MTL architecture, where the skip connections are from the reconstruction decoder to the segmentation decoder, instead of from the shared encoder to the segmentation decoder. However, the importance of this change is only verbally motivated, but not experimentally demonstrated. I think it would be important to compare against a 'naive' MTL architecture, with a shared encoder and skip connections from the encoder to both decoders.

* I am a bit confused as to what is the main goal of the paper. Is it to obtain segmentations from under-sampled k-space data? Or is it to get both segmentations as well as reconstructions? If it is the former, I think it would make sense to check if a CNN can directly segment the under-sampled image, if trained like this in a supervised manner. If it is the latter, it would be important to compare with at least one of the several methods in the literature proposed for reconstructing under-sampled MRIs.

**Justification Of Rating:**

With a relatively small methodological novelty, I think this is a mainly validation paper.
In this case, appropriate comparisons with related works (as mentioned in the 'weaknesses') are important for acceptance, in my opinion.

**Paper Type:**

validation/application paper

**Special Issue:**

no

---

> ### Author Response · Authors · 2020-03-25
> **Response to Reviewer 3 part 1**
>
> We appreciate the reviewer’s interest in the problem tackled in our paper, especially its potential for the MRI field. We understand concerns about the importance of introducing skip connections between tasks. We guarantee our architectural choices are a result of deep reasoning and experiments.
>
> Q1: It would make sense to check if a CNN can directly segment the under-sampled image.
> A1: We are currently training a network to segment cartilage directly from undersampled MRIs and we will report the results of this experiment when complete.
>
> Q2: It would be important to compare against a 'naive' MTL architecture, with a shared encoder and skip connections from the encoder to both decoders.
> A2: We are currently training a 'naïve' MTL architecture, with a shared encoder and skip connections from the encoder to both decoders and we will report the results of this experiment when complete.
>
> Q3: I am a bit confused as to what is the main goal of the paper.
> A3: We aim to perform segmentation from highly undersampled MRIs. On the other hand, this work demonstrates that performing tasks directly from undersampled MRIs is possible.
> The reason why we believe it is important to have a reconstruction output in addition to the segmentation one (or any other task’s output) is strictly related to the clinical world: it is through the expert knowledge of radiologists that we validate a reconstruction framework, as shown in the NeurIPS 2019 reconstruction challenge. By solely training a network to perform segmentation on undersampled MRIs, the only route available for assessment of segmentation quality is a quantitative evaluation against a ground truth, which may be often lacking in cases where no surgery was performed or if the knee wasn't visualized arthroscopically. Radiologists can evaluate accuracy of both reconstruction and segmentation with their additional access to patient's clinical notes. Moreover, their knowledge of confounders on imaging such as motion or partial volume artifacts can help them assess the reconstructed image better.
>
> Q4: Compare with at least one of the several methods in the literature proposed for reconstructing under-sampled MRIs.
> A4: Although reconstruction is not the main goal of our paper, we compare our reconstruction performance against an L1-Wavelet compressed sensing algorithm (CS)[1,2] and a UNet architecture proposed in [3] as proposed as benchmark method by the FastMRI team.
> Metric: SSIM
> Methods	|	2×	     |         	4×	|	    6×	|	   12×	|	24×
> Proposed	|89.74±0.47|80.06±1.16	|75.84±1.60	|69.05±2.27	|62.25±2.84
> L1-wavelet CS|86.90±0.74 |66.48±2.08	|65.70±2.10	|53.11±2.72	|41.99±2.87
> UNet		|89.23±0.49|79.25±0.17	|74.98±0.61	|67.51±0.32	|59.65±0.81
> Metric: NRMSE
> Methods	|	   2× 	|	    4×	|	    6×	|	   12×	|	24×
> Proposed	|15.80±2.74	|20.87±3.51	|22.46±3.77	|26.06±4.32	|29.6±4.63
> L1-wavelet CS|21.80±0.88	|42.68±2.54	|27.83±0.76	|35.76±1.15	|49.69±1.25
> UNet		|15.90±1.38	|21.89±1.99	|23.70±2.09	|27.47±1.23	|31.69±2.39
>
> L1-Wavelet compressed sensing algorithm provided a very accurate reconstruction at 2x. At higher AFs, the lack of details in the undersampled image negatively affects CS’s performance. This is less of a problem for deep learning methods; through a data driven training procedure, priors- which are instrumental for the reconstruction- are introduced. We provided additional comments on this topic in our response to Reviewer 4 Q3, and would like to kindly invite you to read them.
>
> [1] M. Lustig, D. L. Donoho, J. M. Santos, and J. M. Pauly, “Compressed sensing MRI,” IEEE Signal Process. Mag., vol. 25, no. 2, p. 72, 2008.
> [2] http://indexsmart.mirasmart.com/ISMRM2019/PDFfiles/4819.html
> [3] Jure Zbontar, Florian Knoll, Anuroop Sriram, Matthew J Muckley, Mary Bruno, Aaron De-fazio, Marc Parente, Krzysztof J Geras, Joe Katsnelson, Hersh Chandarana, et al. fastmri:An open dataset and benchmarks for accelerated mri.arXiv preprint arXiv:1811.08839,2018.

---

> > ### Author Response · Authors · 2020-03-28
> > **Response to Reviewer 3 part 2**
> >
> > Q1: It would make sense to check if a CNN can directly segment the under-sampled image.
> > A1: We trained a VNet-like encoder decoder – which like TB-recon is comprised of 2 levels with 1 and 2 conv per level, 3 conv at the bottom level and 16 feature channels at the input layer. The network is trained to only segment zero-filled reconstructed MRIs with no denoising task involved and we refer to it with ‘zero-filled’.
> >
> > Femur - DSC
> > Method                  2x  		4x  	                      6x  			12x 		     24x
> > Proposed       88.08±1.98	87.33±1.93	87.58±1.79	86.97±2.25	85.63±2.56
> > zero-filled	83.73±3.11  	82.27±2.75  	81.79±2.73  	81.60±2.83  	16.27+0.91
> >
> > Tibia - DSC
> > Method              2x  		4x  		                6x  			12x 		    24x
> > Proposed       86.91±3.15  85.12±3.85	86.18±3.59	86.09±3.49	85.84±3.51
> > zero-filled	83.90±4.79  83.32±4.53  	81.26±5.20  	82.17±4.54  	29.75±1.80
> >
> > Patella - DSC
> > Method                   2x  		    4x  		     6x  		   12x 	              24x
> > Proposed        82.17±6.09	82.10±6.83	81.76±7.82	80.67±8.26	77.65±7.96
> > zero-filled   	 80.42±7.56 	78.10±7.17  	77.37±6.47  	76.80±7.65 	70.88±10.59
> >
> > Menisci - DSC
> > Method                    2x  		4x 	               6x  		     12x 			24x
> > Proposed         84.99±3.09	84.91±2.79	83.78±3.06	83.29±3.14	82.28±2.64
> > zero-filled   	  82.38±3.53 	81.87±2.99  	81.61±3.15  	80.68±3.12  	16.69±0.93
> >
> > We believe that this experiment was instrumental to show that image quality is crucial for high quality segmentations. Hence, incorporating a denoising step to the segmentation procedure is necessary to obtain good segmentations. This is especially true at high acceleration factors where in most of the compartments poor segmentations were obtained, mainly because zero-filled MRIs lack of fine details which allow a correct identification of the tissue of interest.
> >
> > Q2: It would be important to compare against a 'naive' MTL architecture, with a shared encoder and skip connections from the encoder to both decoders.
> > A2: We trained a ‘naïve’ architecture, which differs from TB-recon in the way skip connections are implemented, as per the comment received. We report the results obtained at 4x, 6x and 24x. Since the remaining 2 networks have not completed training at the rebuttal submission deadline, we will report results in the amended manuscript.
> >
> > Femur - DSC
> > Method            4x        	            6x          	    24x
> > Proposed      	87.33±1.93  87.58±1.79      85.63±2.56
> > zero-filled   	82.27±2.75  81.79±2.73     16.27+0.91
> > Naive              84.97±2.69  82.94±2.99     83.71±2.82
> >
> > Tibia - DSC
> > Method               4x         		6x          	     24x
> > Proposed       85.12±3.85	86.18±3.59	85.84±3.51
> > zero-filled   	83.32±4.53  	81.26±5.20  	29.75±1.80
> > Naive              84.09±3.86  	83.26±4.20  	56.15±2.38
> >
> > Patella - DSC
> > Method                   4x         	 6x            	24x
> > Proposed       82.10±6.83  81.76±7.82   77.65±7.96
> > zero-filled   	78.10±7.17  77.37±6.47    70.88±10.59
> > Naive              53.74±4.36  77.62±7.61    32.45±2.24
> >
> > Menisci - DSC
> > Method                  4x                    6x                 24x
> > Proposed       84.91±2.79	83.78±3.06	82.28±2.64
> > zero-filled   	81.87±2.99  	81.61±3.15    	16.69±0.93
> > Naive              82.35±2.69  	81.47±3.31    	54.59±2.24
> >
> > We thank the reviewer for the request to conduct this experiment. We believe that the results of this experiment are in line with what was stated in A1. Results show that image segmentation is improved by including the image reconstruction task at all acceleration factors. Furthermore, we argue that TB-recon outperforms the naïve architecture, mainly because of the presence of skip connections between the two decoders. We believe our proposed skip connections are key in introducing to the segmentation task features which are representative of higher resolution images, and yet these features’ quality enhances as the reconstruction improves.

---

> > > ### Comment · AnonReviewer3 · 2020-03-28
> > > **New experiments improve validation study. I would like to improve my rating to "4. strong accept"**
> > >
> > > Thanks for carrying out the suggested experiments. I think they definitely increase the usefulness of your work. It can be seen quite clearly that (1) it is not feasible to segment under-sampled data without an intermediate reconstruction step and (2) the novelty of having skip connections between the decoders is beneficial over having a 'naive' MTL architecture with skip connections between the shared encoder to each decoder.
> > >
> > > I would like to change my rating to "4. strong accept". Provided these experiments and a brief discussion of their results are added to the paper, I see no major weaknesses in the paper.
> > >
> > > As an additional comment, I would advise the authors to reconsider toning down the title of the paper. "Breaking speed limits" sounds like a bit of over-advertising. "ultra-fast" already conveys the point, in my opinion.

---

> > > > ### Author Response · Authors · 2020-04-02
> > > > **Response to Reviewer 3**
> > > >
> > > > We appreciated the constructive feedback provided by the reviewer, as the additional experiments helped further validate our proposed method.
> > > >
> > > > With regard to the current speed limits, we refer to the experiments we conducted as part of our rebuttal, specifically those in response to Q4, where we compare our proposed approach against compressed sensing, a traditional reconstruction method. From the experiments, compressed sensing provided a poor reconstruction beyond 2x AF. We consider this a fair representative of the current speed limit. With respect to deep learning methods, we consider that entries to the fast-MRI challenge in Neurips 2019 [1] are representative of state-of-the-art methods in fast-MRI, and for single coil acquisitions, the AF was 4x. Furthermore, the ultimate goal of the fast-MRI project is to achieve 10x faster scans [2]. However, it is important to remark that we aim to perform segmentation from undersampled 3D knee MRIs. With regard to segmentation-based MRI reconstruction, we cite [3] (other related works have been cited in the manuscript), where the AF is 5x, the approach is 2D and the two tasks are pretrained independently. With that in mind, we would strongly consider modifying the title of the manuscript if this made the paper better received by the community.
> > > >
> > > > We will report the ablation study and additional experiments in the revised version of the manuscript.
> > > >
> > > > Ultimately, we thank the reviewer for improving their score for our paper.
> > > >
> > > > [1] https://fastmri.org/leaderboards/challenge
> > > > [2] https://ai.facebook.com/blog/fastmri-challenge/
> > > > [3] Sun, Liyan, et al. "Joint CS-MRI reconstruction and segmentation with a unified deep network." International Conference on Information Processing in Medical Imaging. Springer, Cham, 2019.

---

### Meta-Review · Area_Chair1 · 2020-03-31
**MetaReview of Paper239 by AreaChair1**

**Rating:** 3
**Recommendation For Accepted Papers:** Oral

**Metareview:**

This paper proposes a multi-task network to further break the speed limits. The idea is fine. However, the main concern I have is regarding the speed limits?　Can you provide us what's the current speed limits  of exisiting methods and what's your breakthrough with the proposed method?  I suggest acceptance but would like to get the feedback from the authors.

**Paper Type:**

both

**Special Issue:**

no

---

> ### Author Response · Authors · 2020-04-02
> **Response to the Area Chair 1**
>
> We thank the Area Chair for recommending our paper to be accepted and for suggesting it for oral presentation.
>
> Q1: Provide us what's the current speed limits  of existing methods?
> A1: With regard to the current speed limits, we refer to the experiments we conducted as part of our rebuttal, specifically those in response to Reviewer 3 Q4 and Reviewer 4 Q2, where we compare our proposed approach against compressed sensing, a traditional reconstruction method. From the experiments, compressed sensing provided a poor reconstruction beyond 2x AF. We consider this a fair representative of the current speed limit. With respect to deep learning methods, we consider that entries to the fast-MRI challenge in Neurips 2019 [1] are representative of state-of-the-art methods in fast-MRI, and for single coil acquisitions, the AF was 4x. Furthermore, the ultimate goal of the fast-MRI project is to achieve 10x faster scans [2].
> However, it is important to remark that we aim to perform segmentation from undersampled 3D knee MRIs. With regard to segmentation-based MRI reconstruction, we cite [3] (other related works have been cited in the manuscript), where the AF is 5x, the approach is 2D and the two tasks are pretrained independently.
>
> Q2: what's your breakthrough with the proposed method
> A2: Our contributions to the field are:
> - We propose an encoder-multi-decoder network, with skip connections among the two decoders. With our ablation study – which we reported in the response to Reviewer 2 Q1 – we demonstrated that our architectural choice improved segmentation performance.
> - Our approach is 3D, and both tasks are trained end-to-end simultaneously. The method - which starts from undersampled MRI data and train on both image denoising as well as on tissue segmentation – shows great potential for additional improvements in the acceleration of MRI scans.
> - We demonstrated that it is possible to perform tissue segmentation directly from heavily undersampled k-space (up to 24x AF) and obtain quality results comparable to those in fully-sampled scenarios as shown in the response to Reviewer 2 Q3.
>
> [1] https://fastmri.org/leaderboards/challenge
> [2] https://ai.facebook.com/blog/fastmri-challenge/
> [3] Sun, Liyan, et al. "Joint CS-MRI reconstruction and segmentation with a unified deep network." International Conference on Information Processing in Medical Imaging. Springer, Cham, 2019.

---

### Decision · Program_Chairs · 2020-04-11

Accept